# Modeling the spatial dependence of floods using the Fisher copula

Manuela I. Brunner[1,2,3], Reinhard Furrer[4], and Anne-Catherine Favre[2]

[1]Department of Geography, University of Zurich, Zurich, Switzerland
[2]Univ. Grenoble Alpes, CNRS, IRD, Grenoble INP, IGE, Grenoble, France
[3]Swiss Federal Institute for Forest, Snow and Landscape Research WSL, Birmensdorf ZH, Switzerland
[4]Department of Mathematics and Department of Computational Science, University of Zurich, Zurich, Switzerland

**Correspondence:** Manuela Brunner (manuela.brunner@wsl.ch)

**Abstract.** Floods do often not only affect a single location but a whole region. Flood frequency analysis should therefore be undertaken at a regional scale which requires the considerations of the dependence of events at different locations. This dependence is often neglected even though its consideration is essential to derive reliable flood estimates. A model used in regional multivariate frequency analysis should ideally consider the dependence of events at multiple sites which might show dependence in the lower and/or upper tail of the distribution. We here seek at proposing a simple model that on the one hand considers this dependence with respect to the network structure of the region and on the other hand, allows for the simulation of stochastic event sets at both gauged and ungauged locations. The new Fisher copula model is used for representing the spatial dependence of flood events in the nested Thur catchment in Switzerland. Flood event samples generated for the gauged stations using the Fisher copula are compared to samples generated by other dependence models allowing for modeling multivariate data including elliptical copulas, R-vine copulas, and max-stable models. The comparison of the dependence structures of the generated samples shows that the Fisher copula is a suitable model for capturing the spatial dependence in the data. We therefore use the copula in a way such that it can be used in an interpolation context to simulate event sets comprising gauged and ungauged locations. The spatial event sets generated using the Fisher copula well capture the general dependence structure in the data and the upper tail dependence, which is of particular interest when looking at extreme flood events and when extrapolating to higher return periods. The Fisher copula was for a medium-sized catchment found to be a suitable model for the stochastic simulation of flood event sets at multiple gauged and ungauged locations.

## 1 Introduction

Reliable flood estimates are needed to protect settlements and infrastructure against future floods. Such estimates have traditionally been derived by looking at runoff observations at a single measurement station (Neal et al., 2013). Severe precipitation events, however, do often not only affect a sub-catchment but several ones. This results in widespread floods whose probability is of interest for flood mitigation and for assessing financial risk in the reinsurance industry (Keef et al., 2013). To estimate the probability of widespread floods, a regional multivariate frequency analysis is required which jointly considers discharge values at multiple locations (Serinaldi and Kilsby, 2016). Such an analysis must take into account the dependence of events in different sub-catchments (Thibaud et al., 2013). Spatial dependence was found to be important because no single event caused the maximum flood extent at all locations and assuming perfect correlation between tributaries overestimated flood hazard

(Neal et al., 2013). A multivariate approach is therefore needed which represents the spatial dependence between floods from different watersheds (Schulte and Schumann, 2015) which might be radially asymmetric and show non-null tail dependence since more extreme events might be more strongly related than less extreme events. A multivariate approach should allow for the generation of stochastic event sets at multiple stations and for the consideration of the network structure of the catchment.

Such event sets can be used for the estimation of rare events which are missing in the observed records and which affect a larger area. In addition to the generation of event sets at gauged stations, the approach should also allow for the simulation of stochastic event sets at ungauged locations since flood estimates might be required at locations where no runoff observations are available. Both the fitting of the models and the interpolation process should be simple such that they can be easily interpreted and be applied by practitioners.

The spatial dependence of flood events at several locations has been assessed using different types of approaches comprising conditional exceedance models (Keef et al., 2013; Neal et al., 2013), hierarchical Bayesian models (Yan and Moradkhani, 2015), the mulivariate skew-$t$ distribution (Ghizzoni et al., 2010, 2012), max-stable models (Wang et al., 2014), and copula models such as pair-copula constructions (Bevacqua et al., 2017; Gräler, 2014; Schulte and Schumann, 2015), hierarchical Kendall copulas (Fischer et al., 2017), and factor copula models (Lee and Joe, 2017). Max-stable and copula models are

two classes of approaches which are well established in multivariate extreme value modeling and have been used in other contexts than those of floods. Max-stable distributions arise out of the study of the limiting behavior of vectors of component-wise maxima. There exists a number of parametric max-stable models (Segers, 2012). The consideration of concomitance in annual maxima, however, only informs on whether maxima tend to co-occur in the same year without accounting for their timing within a year (Blanchet et al., 2018). To overcome this, the max-stable framework has been extended to the peak-over-

threshold setting (Thibaud et al., 2013). Max-stable models have often been applied in the context of extreme precipitation (see e.g. Blanchet et al., 2018; Davison et al., 2012; Stephenson et al., 2016) and less in the context of floods since most models presupposed that extremal dependence depends only on Euclidean distance (Asadi et al., 2015). While max-stable models describe the marginal distribution and the dependence structure at the same time, copula models allow for the modeling of the dependence structure of multivariate distributions separately from their univariate marginals (Genest and Favre, 2007).

Copula based models have also been found to be well suited for the spatial interpolation of extremes (Gräler, 2014) since they overcome the limitations of the classical variogram which is sensitive to outliers and influenced by the marginal distribution of the observations (Kazianka and Pilz, 2010). Copula models are therefore a flexible alternative to max-stable models. Bárdossy and Li (2008) have introduced an interpolation method based on copulas. They have used both the Gaussian and a $v$-transformed normal copula, which was used by Durocher et al. (2016) for predicting flood quantiles at ungauged basins

based on physiographical locations. Quessy et al. (2016) applied the general class of chi-square copulas in a spatial context. The family of Fisher copulas generalizes the class of centered chi-square dependence models (Favre et al., 2018). Gräler (2014) introduced spatial vine copulas which are a combination of bivariate copulas not limited to a single copula family. Because of that, they allow for both varying strength of dependence and a changing dependence structure with distance. However, the use of spatial vine copulas requires fitting several models and makes interpretation more difficult since subsequent pair-copulas are

conditioned on previous ones.

The models mentioned above usually do not fulfill all of the requirements to satisfactorily generate stochastic flood event sets in a nested catchment with gauged and ungauged locations. Max-stable models do usually not allow for the consideration of the network structure. An exception is the model of Asadi et al. (2015) which distinguishes between flow-connected and flow-unconnected stations. Conditional exceedance models and spatial vine copulas require the fitting of several models. Classical copula models do usually not allow for spatial interpolation and those which allow for interpolation do not allow for both radial asymmetry and non-null tail dependence. The normal copula has well established theoretical properties but it can only represent dependence structures that are radially symmetric and does not incorporate tail dependence. The Student copula and other elliptical copulas allow for tail dependence but are also radially symmetric (Favre et al., 2018). The chi-square copulas introduced by Bárdossy (2006) are radially asymmetric but their tail dependence coefficients are null. Favre et al. (2018) have shown that the limitations of these copulas in terms of tail dependence and asymmetry can be overcome by the recently introduced Fisher copula. It allows for high dimensional modeling, upper tail dependence, and radial asymmetry and was therefore found to be well suited to model multi-site precipitation data in Switzerland. So far, the Fisher copula has only been used in a simulation context for generating stochastic precipitation event sets at gauged locations. However, it has not yet been used in a spatial context to interpolate to ungauged locations.

In this study, we aim at using the Fisher copula such that it can be applied not only to generate stochastic flood event sets at multiple locations but also in an interpolation context for the generation of event sets including ungauged locations. More specifically, we address the following research questions:

- What is a suitable distance measure for explaining the correlation of events at different locations within a nested catchment?

- Can the Fisher copula capture the dependence structure of the floods observed at several locations within a nested catchment and what is its performance compared to other dependence models in use?

- How can the Fisher copula be used to generate event sets for multiple locations including ungauged locations?

These questions are answered by applying several dependence models in the nested Thur catchment in Switzerland which is described in the next section. Section 3 then describes how a synchronous flood event set is sampled from observed runoff time series, how the marginal behavior and the dependence structure of the flood sample is modeled, and how the Fisher copula is extended to simulate floods at ungauged locations. In Section 4, we compare the dependence structure and the marginal behavior of event sets simulated by the Fisher copula to those of event sets generated by other types of dependence models including two max-stable models and several copula models including elliptical copulas, an extreme value copula, and the R-vine copula. Furthermore, the dependence structure of event sets including ungauged locations is analyzed. Section 5 discusses limitations and perspectives of the proposed approach.

## 2  Study area

This study was conducted using flood samples from ten gauging stations situated in the nested Thur catchment in northeastern
Switzerland (Figure 1). The Thur catchment is a tributary of the Rhine and does neither have considerable retention areas
(Guldener and Wieland, 1980) nor is it strongly regulated (Girons Lopez and Seibert, 2016). The catchment has an area of
5   1696 km$^2$ and its elevation ranges from 356 to 2503 m.a.s.l. with a mean elevation of 770 m.a.s.l. (Federal Office for the
Environment FOEN, 2009). The flow regime is snow melt dominated and the average rainfall is 1350 mm. This rainfall is
distributed over all seasons with the highest amounts in summer. Large precipitation events in the headwaters might cause a
rapid discharge build-up in the basin due to the steep terrain and short concentration times (Girons Lopez and Seibert, 2016).
The ten gauging stations are distributed over the whole catchment area and subdivide the catchment into ten sub catchments.
10  These are listed in Table 1 together with their main catchment characteristics and an abbreviation that will be used in the sequel
of this paper to refer to the individual stations. For each of the measurement stations, an hourly runoff time series spanning 30
to 40 years was available from the Federal Office for the Environment (FOEN) and used as the basis for flood sampling.

**Table 1.** Catchments used for fitting the spatial dependence models tested in this study. The station abbreviation is given together with the
full name of the river and the gauging station, the catchment area, the station elevation, the mean elevation of the catchment, and the record
period.

| Abbre-viation | River | Station name | Catchment area [km$^2$] | Station elevation [m.a.s.l] | Mean elevation [m.a.s.l] | Record [from-to] |
|---|---|---|---|---|---|---|
| ThuSte | Thur | Stein | 84 | 850 | 1448 | 1974-2004 |
| NecMog | Necker | Mogelsberg | 88 | 606 | 959 | 1974-2013 |
| ThuJon | Thur | Jonschwil | 493 | 534 | 1030 | 1974-2013 |
| GlaHer | Glatt | Herisau | 16 | 679 | 840 | 1974-2013 |
| SitApp | Sitter | Appenzell | 73 | 790 | 1186 | 1974-2013 |
| SitStg | Sitter | St.Gallen | 256 | 636 | 975 | 1980-2008 |
| ThuHal | Thur | Halden | 1085 | 456 | 910 | 1974-2013 |
| MurWae | Murg | Wängi | 79 | 466 | 650 | 1974-2013 |
| MurFra | Murg | Frauenfeld | 212 | 390 | 580 | 1974-2013 |
| ThuAnd | Thur | Andelfingen | 1696 | 356 | 770 | 1974-2013 |

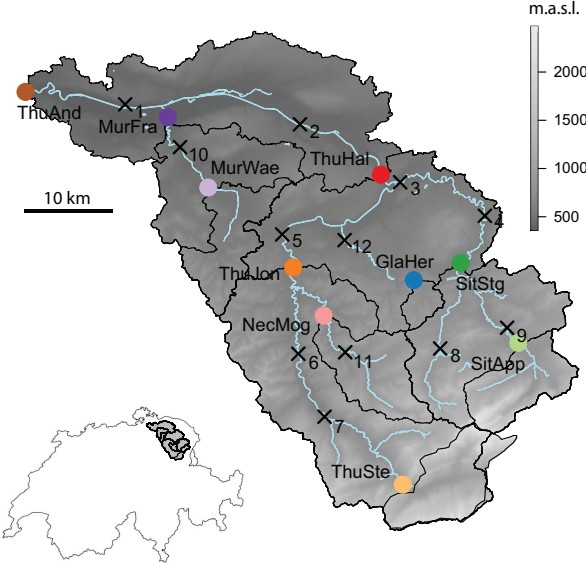

**Figure 1.** Gauging stations within the Thur catchment in northeastern Switzerland (colored dots) and stations used for spatial interpolation (crosses).

## 3 Methods

### 3.1 Event definition

The multivariate analysis was based on a set of flood events which were relevant at the regional scale since a univariate analysis would be sufficient for events relevant only at a local scale. These regional flood events were identified via the three

step procedure illustrated in Figure 2. First, flood events were identified at a local scale for each individual station using a peak-over-threshold approach with the 0.9975 quantile of the individual station as a threshold. The dates of occurrence of these local events were listed. Within all the dates of occurrence, independent events were identified by allowing for only one event per week. Second, a synchronous event set was composed by identifying the maximum event magnitudes corresponding to these independent events for each of the individual stations. This procedure allowed for the composition of an event set with events during which at least one station exceeded its 0.9975 quantile. This event set represents an upper set of events that are

considered to be dangerous and therefore represent a hazard scenario as defined by Salvadori et al. (2016). However, this set consisted of both events that were only relevant at a local scale where univariate frequency analysis is sufficient and events that were relevant at a regional scale where the consideration of the dependence between events is relevant. Third, these regional events were therefore separated from the local events. To qualify for a regional event, where several stations are jointly affected by a flood event, two criteria had to be fulfilled: 1) the event had to be of similar importance at the individual stations and 2) over

all stations, the event had to belong to the most important ones. The event highlighted in step 3 of Figure 2 would for example be

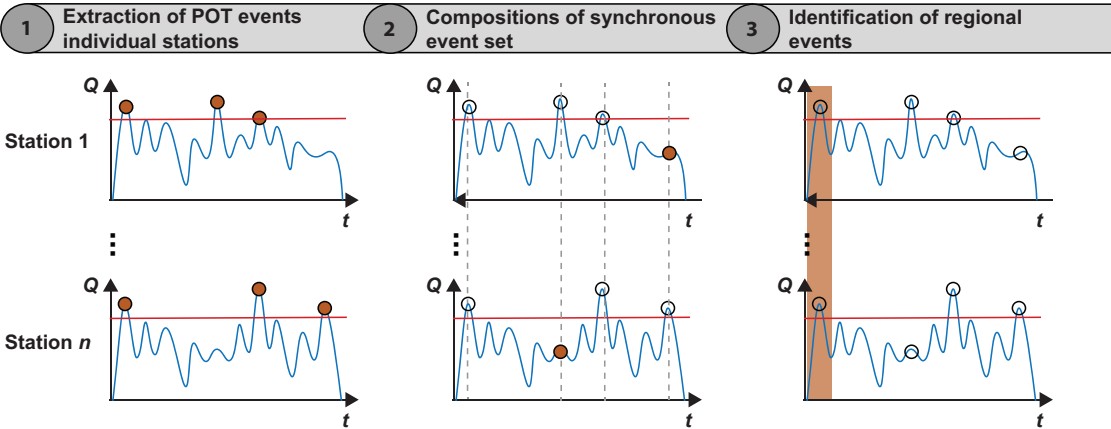

**Figure 2.** Illustration of the flood event identification procedure consisting of three main steps: 1) Extraction of peak-over-threshold (POT) events at individual stations, 2) composition of synchronous event set, and 3) identification of regional events within the synchronous event set.

chosen because if fulfills both criteria. In contrast, the third event might not be chosen because it was not of similar importance at the individual stations. These criteria were quantified in terms of the ranks of the events within the individual station series in order to make observations in catchments with a different size and therefore different event magnitudes comparable. An event could have a low rank in one series but a high rank in another series, which would lead to a high variability in ranks

across stations. On the contrary, an individual event could be assigned similar ranks at different stations, which would lead to a low variability in ranks across stations. Criterion one was a low variability of the ranks of a specific event across different stations (standard deviation of ranks < 50) and criterion two a high rank sum over all stations (rank sum > 1500). The marginal distributions of local and regional events were compared to check whether they were actually distinct. The location parameter was higher for the regional than for the local events and the shape parameters of the two event sets were also clearly distinct.

The shape parameter of local events showed a low mean and a high variability across stations while the regional events showed a higher mean and low variability. The spatial dependence of events assessed via Kendall's tau was higher for the regional than for the local events. The local events were found to be not necessarily regionally important and were therefore excluded from subsequent analyses. The 63 regional events identified occurred in all seasons with a concentration in summer. The regional event set is subsequently referred to as the event set.

## 3.2 Spatial dependence

The spatial dependence of events was assessed via Kendall's tau for all pairs of stations (Figure 3). It shows that there is a generally positive dependence between events at most stations while the dependence is strongest at stations that are closely linked along the stream (e.g. MurWae and MurFra or ThuHal and ThuAnd). Both upper and lower tail dependence were present in the data according to the estimator of Schmidt and Stadtmüller (2006) which needs to be used with care since it provides

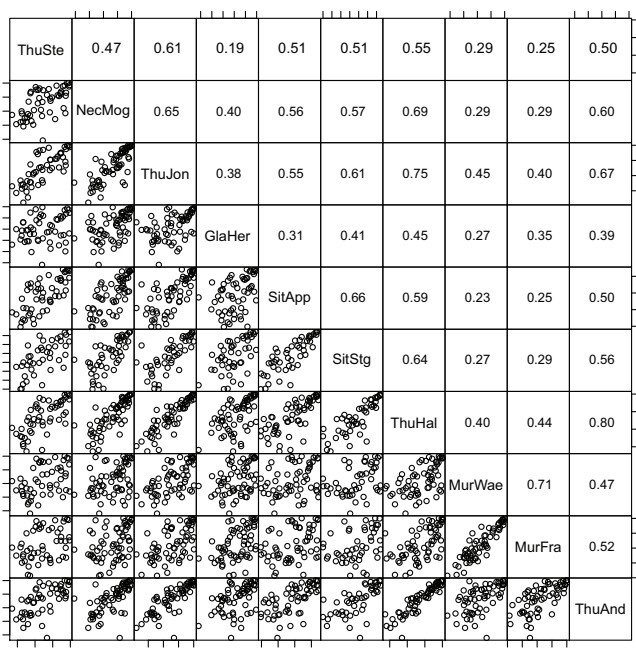

**Figure 3.** Pairs plot (lower triangular matrix) and Kendall's correlation coefficients (upper triangular matrix) of regional peak-over-threshold events. The stations are labeled with their abbreviations listed in Table 1. Patterns of positive association are visible in all cases.

unreliable estimates for small sample sizes (Serinaldi, 2015). However, upper tail dependence can also be assumed to be present since extreme precipitation events, which might cause extreme flood events, have been shown to exhibit upper tail dependence (see e.g. Evin et al., 2018; Naveau et al., 2016).

Spatial methods often relate the dependence between events at two stations to the distance between these stations. Traditionally, the Euclidean distance has been used to do so, which might not be very relevant in the case of floods since they evolve along a river network. A comparison of different distance measures (Euclidean, river distance, physiographical distance) showed that river distance and distance regarding mean catchment elevation explained Kendall's tau best. We used river distance as a distance measure since it has a hydrological meaning. The R-package *riverdist* (Tyers, 2017) was used to compute river distance based on a line shapefile of the river network.

A model depicting the behavior of joint events at multiple stations needs to account for both the marginal behavior of the variables and their dependence structure. These two elements are addressed in turn.

### 3.3 Marginal model

The marginal distribution of extreme values is usually modeled using a GEV distribution in an annual maxima framework or a generalized Pareto distribution (GPD) in a peak-over-threshold framework (Coles, 2001). Strictly speaking, we were in neither of the two frameworks since we were working with a synchronous dataset that has been composed of peak-over-threshold events at individual stations and complemented with the corresponding events at the remaining stations. The event sets at the

individual stations were well fitted by the GEV distribution. The generalized extreme value distribution (GEV) (Coles, 2001) was not rejected for both types of events in most catchments according to the Anderson-Darling test statistic computed using the procedure proposed by Chen and Balakrishnan (1995) (level $\alpha = 0.05$). This statistic was introduced for testing the validity of skewed distributions when unknown parameters must be estimated from the sample data. The marginal GEV distributions were fitted for each station separately using maximum likelihood estimation. A regional estimation of the shape parameter $\xi$ would have been desirable (Evin et al., 2016). Pooling the events of different stations, however, was not sensible since even the specific discharges were dependent on the catchment area. A spatial model of the marginal distributions was derived using trend surfaces (Davison et al., 2012). In a first step, the location and scale parameters were normalized by catchment area. In a second step, a regression model was fitted. for each parameter of the GEV distribution which allowed for predictions at a given location using catchment characteristics only. Suitable models were identified using stepwise backward regression (Harrell, 2015) on a set of 53 catchment characteristics (for a list and a short description see Table A1 in Viviroli et al., 2009). The final linear models used the following explanatory variables:

- Location parameter: $X$-coordinate, percentage area of hard rock, variability of the Julian Date of the annual maximum hourly precipitation.

- Scale parameter: $X$-coordinate, $Y$-coordinate, catchment area, catchment elevation.

- Shape parameter: catchment shape, soil topographic index, percentage area of hardrock, pasture and arable land in contributing areas, urban area in contributing areas.

They resulted in absolute prediction errors over the ten stations of 0.11, 0.21, and 0.15 respectively i.e., already the use of the spatial marginal model leads to prediction errors, which is, however, not the focus of this study.

## 3.4 Dependence model

The dependence structure was treated independently of the marginal distributions. We modeled the spatial dependence of the flood events using two types of approaches: copula-based approaches and max-stable approaches. While the copula approach assumes uniform marginals, max-stable process models assume unit Fréchet marginals (Cooley et al., 2012). Although the copula and the extreme value frameworks share some connections, only a few authors from the extreme value community adopt the copula framework for multivariate extremes (Ribatet and Sedki, 2013). The different approaches are summarized in Table 2 and described in the paragraphs below.

### 3.4.1 Copula models

The copula approach has its origin in the representation theorem of Sklar (1959) which states that the joint cumulative distribution function $F_{XY}(x,y)$ of any pair of continuous random variables $(X, Y)$ can be written as:

$$F_{XY}(x,y) = C\{F_X(x), F_Y(y)\}, x, y \in \mathbb{R}, \tag{1}$$

where $F_X(x)$ and $F_Y(y)$ are the marginal distributions and $C : [0,1]^2$ is the copula. $C$ is unique if the marginals are continuous. One of the main advantages of the copula approach is that the selection of an appropriate copula for modeling the dependence between $X$ and $Y$ can proceed independently from the choice of their marginal distributions (Genest and Favre, 2007). An important property of a copula is whether the dependence is the same for high and low values (Bárdossy and Li, 2008) which implies tail symmetry. For a theoretical introduction to copulas, the reader is referred to Durante and Sempi (2015); Joe (2014) or Nelsen (2005) and for an introduction with an engineering focus to Genest and Favre (2007); Salvadori et al. (2007) and Salvadori and De Michele (2007). Guidelines for using copulas in a hydrological and natural hazard context are provided in Favre et al. (2004); Salvadori and De Michele (2004); Salvadori et al. (2014, 2015) and Salvadori et al. (2016).

We tested five different types of copulas, which all allow for modeling in high dimensions and therefore at multiple sites: 1) the Gaussian copula, 2) the Student-$t$ copula, 3) the Gumbel copula, 4) the R-vine copula, and 5) the recently developed Fisher copula.

**Table 2.** Modeling capabilities of copula and max-stable models tested to model spatial dependence. Asymmetry and tail dependence are not specified in the context of max-stable models.

| Model | Asymmetry | Upper tail dependence | Lower tail dependence | Asymptotic dependence | Simulation at multiple stations | Spatial interpolation | Simple model setup |
|---|---|---|---|---|---|---|---|
| Gaussian copula | No | No | No | No | Yes | Yes | Yes |
| Student-$t$ copula | No | Yes | Yes | No | Yes | Yes | Yes |
| Gumbel copula | Yes | Yes | No | No | Yes | No | Yes |
| R-vine copula | Yes | Yes | Yes | No | Yes | Yes | No |
| Fisher copula | Yes | Yes | No | No | Yes | Yes | Yes |
| Gaussian max-stable | - | - | - | Yes | Yes | Yes | Yes |
| Brown-Resnick max-stable | - | - | - | Yes | Yes | Yes | Yes |

1. Gaussian copula: The Gaussian copula is completely determined by the knowledge of the correlation matrix $\Sigma$ and the parameters of the Gaussian copula are simple to estimate. However, the Gaussian copula does not have tail dependence (Malevergne and Sornette, 2003) and the dependence is symmetrical (Bárdossy and Li, 2008).

2. Student-$t$ copula: The description of a Student-$t$ copula relies on two parameters: the correlation matrix $\Sigma$ and the number of degrees of freedom $\nu$. In contrast to the Gaussian copula, it allows for tail dependence (Malevergne and Sornette, 2003) but the dependence is symmetrical (Bárdossy and Li, 2008). Both the Gaussian and Student-$t$ copulas allow for the description of spatial variability and therefore for the interpolation to ungauged locations.

3. Gumbel copula: The Gumbel copula belongs both to the class of Archimedean and extreme value copulas. It allows for upper tail dependence (Poulin et al., 2007).

4. R-vine copula: Multivariate data can be modeled using a cascade of simple building blocks, called pair-copulas, in a flexible way (Aas et al., 2009). There is an enormous number of possible R-vine tree sequences to choose from. Dißmann et al. (2013) therefore proposed an automated model selection and estimation technique which is implemented in the R-package *VineCopula* (Schepsmeier et al., 2017).

5. Fisher copula: The Fisher copula arises when components of a Student random vector are squared. This construction procedure follows the construction of the chi-square copula family which is obtained when squaring the components of a normal random vector (Bardossy, 2007). The Fisher copula is characterized by the two parameters $\nu$ and $\Sigma$, similar to the Student-$t$ copula. The Fisher copula generalizes the class of centered chi-square dependence models since it tends to the centered chi-square copula as $\nu \to \infty$. For $\boldsymbol{\epsilon} = (\epsilon_1, \ldots, \epsilon_d)^\mathsf{T}$, the $d$-dimensional Fisher copula can be expressed by

$$C_{\Sigma,\nu}^{\mathrm{F}}(u_1, \ldots, u_d) = \sum_{\boldsymbol{\epsilon} \in \{-1,1\}^d} \left( \prod_{j=1}^{d} \epsilon_j \right) C_{\Sigma,\nu}^{\mathrm{t}} \left( \frac{1 + \epsilon_1 u_1}{2}, \ldots, \frac{1 + \epsilon_d u_d}{2} \right), \tag{2}$$

where $C_{\Sigma,\nu}^{\mathrm{t}}$ is the Student copula and $u$ are the uniform marginals. The elements of the parameter take either the value -1 or 1. The parameter does not have a particular role with respect to the dependence structure (Quessy et al., 2016) The Fisher copula allows for modeling at multiple sites, non-vanishing upper tail dependence, radial asymmetry, and a pairwise structure that helps to interpret results (Favre et al., 2018).

The Gaussian, Student-$t$, and Gumbel copulas were fitted using maximum pseudo-likelihood estimation on pseudo-observations. The R-vine copula was estimated using the automated model selection and estimation technique by (Dißmann et al., 2013). The Fisher copula was estimated using the two-step pseudo maximum likelihood estimator proposed by Favre et al. (2018), which allows for estimating the parameters of the Fisher copula ($\Sigma$ and $\nu$) based on the relationship between each entry of $\Sigma$ and the pairwise values of Kendall's tau (Favre et al., 2018).

### 3.4.2 Max-stable models

Max-stable processes extend extremal models to the spatial context. Various parametric models of max-stable processes have been proposed in the literature (Blanchet et al., 2018), including the Gaussian extreme value (Smith et al., 1990) and the Brown-Resnick process (Kabluchko et al., 2009), which were found to fit precipitation data better than other max-stable processes (Davison et al., 2012). Max-stable processes assume asymptotic dependence (i.e. dependence will not disappear at very large distances between stations) and usually use variables converted to Fréchet margins. For a detailed overview on max-stable processes, the reader is referred to Padoan (2013). Standard procedures for fitting max-stable processes were established for coordinates and Euclidean distances. We therefore used classical multi-dimensional scaling (Borg and Groenen, 2010) to search for coordinates which could correspond to the river distance matrix. The max-stable models were then fitted using the Euclidean distances of the coordinates obtained by multidimensional scaling.

### 3.5 Simulation for gauged locations

Each of the dependence models outlined above can be used to generate a random flood event set for the gauged stations in the region under study. We used each of these models to simulate $n = 1000$ flood event sets. These event sets consisted of variables with uniform margins in the case of copula models and of variables with Fréchet margins in the case of max-stable models. To get values on the original scale, these values had to be back-transformed using a probability integral transform (Genest and Rivest, 2001). The back-transformation was achieved by using the predicted parameters of the GEV distribution obtained by the marginal model (see Section 3.3).

### 3.6 Validation

The multivariate distribution per se could not directly be validated since no quantitative goodness of fit test is available to help in rejecting unsuitable models (Ghizzoni et al., 2010). There are a few validation techniques which are called multivariate but they are essentially aggregation metrics (see e.g. Li and Lu, 2018) and therefore not truly multivariate. Each of the samples obtained by simulation was therefore compared to the observations with respect to their marginal behavior and with respect to the dependence structure separately. For the assessment of the dependence structure, we computed different dependence measures such as the F-madogram which is closely related to the extremal coefficient and usually used in the context of max-stable processes, tail dependence coefficients which are usually computed in the context of copulas, and Kendall's tau which tells us something about the general dependence structure of the data. For the assessment of the marginal behavior, we displayed QQ-plots.

The F-madogram summarizes the spatial dependence structure of the data (Cooley et al., 2006) and is expressed as:

$$v^F(h) = \frac{1}{2}\mathbb{E}|F[Z(x+h)] - F[Z(x)]|, \tag{3}$$

where the margin of $Z(x)$ equals Fréchet margins, $F(x) = \exp(-1/x)$, and $h$ is the distance between a pair of stations.

The F-madograms computed for the different simulated samples were compared to the F-madogram of the observations. This allowed for the evaluation of whether the dependence in relation to the distance between stations was captured by the models. Tail dependence describes the dependence in the upper or lower tail of a distribution and the upper tail dependence coefficient describes the probability that one margin exceeds a high threshold given that the other margin also exceeds a high threshold (Poulin et al., 2007). We used the tail dependence estimator of Schmidt and Stadtmüller (2006) with a parameter $\kappa = 5$ to produce upper tail dependence plots for the simulated samples and the observations. Since tail dependence estimators react quite sensitively to small sample sizes (Serinaldi, 2015), we also computed Kendall's tau which is a non-parametric measure of dependence and does not only focus on the tails of the distributions (Poulin et al., 2007).

### 3.7 Interpolation to ungauged location(s)

The model found to be most suitable for modeling the spatial dependence in the regional flood data was used for the simulation of event sets including ungauged locations. The results shown in Section 4 (Figures 7 and 8) indicate that the Fisher copula best

reproduces the dependence structure in the data. The Fisher copula is non-parametric and uses an empirical correlation matrix (Favre et al., 2018). Each entry of the correlation matrix $\Sigma$ can be related to the value of Kendall's tau of a corresponding data pair. The Fisher copula can be adapted to a spatial context since pairwise distances between stations can be incorporated into the model.

Using only the gauged stations in the study catchment, the correlation matrix $\Sigma$ had a dimension of $10 \times 10$. If a new, ungauged location was added, it was converted to a $11 \times 11$ matrix, if $s$ ungauged locations were added to a $(10+s) \times (10+s)$ matrix. The new matrix still consisted of the former $10 \times 10$ matrix in the upper left corner completed with $s$ vectors of correlation values for the new stations with respect to the other stations. The vectors of correlations could be derived via an empirical correlogram. The empirical correlogram consisted of pairwise correlations plotted against the river distance between stations. To derive the values for an ungauged location, its distance to all gauged stations was determined. The correlogram could then be used to derive the correlations corresponding to these distances. These were used to extend the matrix $\Sigma$. The new extended matrix could finally be used as a parameter of the Fisher copula to generate a flood event set for the gauged and ungauged locations in the dataset. The individual steps of this procedure are outlined in more detail below.

1. An empirical correlogram was set up which relates the entries of the $\Sigma$ matrix corresponding to pairs of stations to the distance between these locations (Figure 4 black points).

2. A non-parametric or parametric model for the correlogram was chosen. The empirical correlogram was on the one hand smoothed using smoothing splines and on the other hand fitted by an exponential model. Both models indicated a gradual decrease of correlation with increasing distance if the smoothing parameter of the spline was chosen high enough. Subsequently, we used the exponential model, however, the procedure could also be applied using the smoothing spline.

3. One or several ungauged locations of interest had to be identified. We here used 12 ungauged locations distributed along the river network which could be prone to flooding.

4. For each of these locations, the river distance with respect to all other locations was determined. These values were used as the new entries of the extended $\Sigma$ matrix.

5. The extended matrix was then used to simulate $n = 1000$ random flood event sets for a set of stations comprising gauged and ungauged locations using the new $\Sigma$ as a parameter in the Fisher copula. The second parameter of the Fisher copula $\nu$, which corresponds to the number of degrees of freedom of the Student distribution and is by the Fisher copula required to be an integer, was found to be only weakly sensitive to an extension of $\Sigma$. This was assessed by reestimating $\nu$ using the new simulated data. The estimated $\nu$ remained at $\nu = 7$ when adding only one station and slightly increased for $s > 10$ to $\nu = 8$.

The adaptation of the Fisher copula to the spatial context was validated by using leave-one-out cross-validation. To do so, one of the 10 gauged stations was considered ungauged, and a Fisher copula fit to this reduced dataset. The Sigma matrix was then completed by applying the procedure described above to the station that was considered to be ungauged. The completed matrix

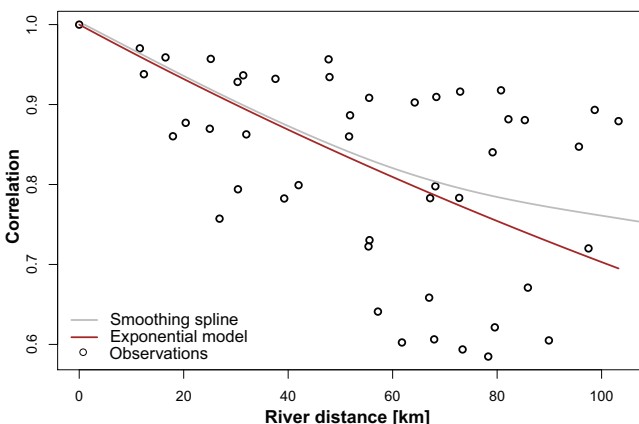

**Figure 4.** Correlation (entries of the $\Sigma$ matrix) plotted against river distance. The observations (dots) are given together with a smoothing spline (grey line), and an exponential model (brown line).

was used to simulate $n = 1000$ event sets which were visually summarized by a Kendall's correlation matrix. A comparison of the correlation matrices resulting by excluding one station at a time with the correlation matrix of the observations showed that these were rather similar.

## 4 Results

### 4.1 Spatial dependence

The exploratory data analysis of the observed flood event sets showed that spatial dependence was present in the data. This is illustrated in Figure 5 by the weekly occurrence of the annual maxima events and the ranks of their magnitudes. A correspondence in the week of occurrence indicates temporal dependence and a similarity in the ranks of the magnitudes across stations indicates spatial dependence. Very severe events were recorded at all stations within the nested catchment and were ranked high in most catchments (e.g. events 1977, 1978, 1999, and 2013). This implies dependence in the upper tail of the multivariate distribution. The most severe events were summer events (see week of occurrence). The ranks of rather weak events were also related (see event 2003 which was a very dry year), which indicates lower tail dependence in the multivariate distribution.

### 4.2 Marginal model

Figure 6 shows how well the marginal distributions of the flood events at the individual stations were modeled by a fitted GEV distribution (see column "Fitted marginals"), by a GEV model with regionalized parameters (see column "Regionalized marginals"), and by samples generated using several dependence models and back transformed using the regionalized GEV model (see columns three to nine). The $p$-values computed by the Kolmogorov-Smirnov test using the individual samples and the observations are provided in Table 3. The marginal distributions of the observations were accurately modeled by the fitted

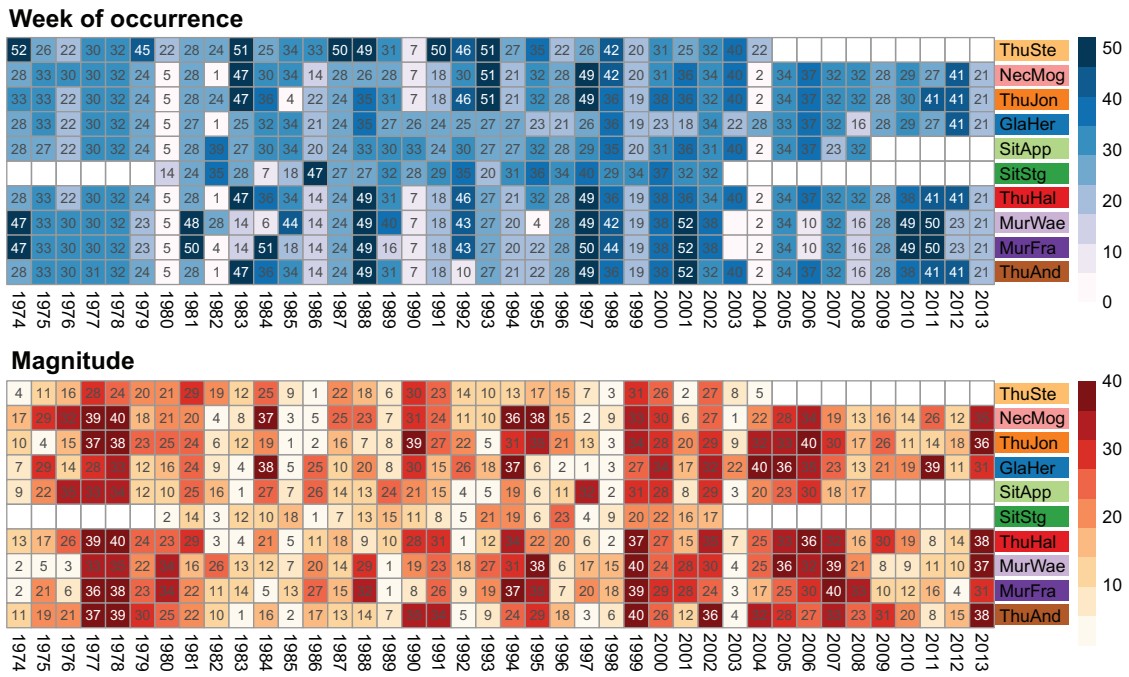

**Figure 5.** Heatplots of the week of occurrence (upper panel) and the magnitude (lower panel) of annual maxima of the ten stations within the study area. The darker the color, the higher the ranks. Missing records lead to empty, white cells.

GEV model. As expected, the GEV model using the regionalized GEV parameters introduced some bias, mainly in the upper part of the distribution. The samples generated using the dependence models and back transformed using the regionalized GEV parameters showed a very similar picture to these regionalized marginal distributions.

## 4.3 Dependence model

Figure 7 shows the F-madogram of the observations and the different dependence models. It shows that the Gaussian, Student-$t$, and the Gumbel copula introduced dependence structures that did not reflect the decreasing dependence among pairs of stations with increasing distance (virtually constant estimates). The Gaussian max-stable model did introduce dependence decreasing with the distance among pairs of stations but this decrease was too strong. The Brown-Resnick max-stable model and the R-vine and Fisher copulas introduced a dependence pattern that closely resembled the one of the observations. The results of the F-madogram were confirmed by the Kendall's tau matrices (Figure 8) which showed that the general dependence structure in the data was best reproduced by the the R-vine and Fisher copulas while the Gaussian, Student-t and Gumbel copula and the Brown-Resnick max-stable models introduced too uniform dependence structures.

Figure 9 shows how well the upper tail dependence in the data was reproduced by the different dependence models. The tail dependence was rather underestimated by the Gaussian and R-vine copulas, while it was overestimated by the Student-$t$ and Gumbel copulas and the Gaussian max-stable model. The Fisher copula model generated samples with variable tail

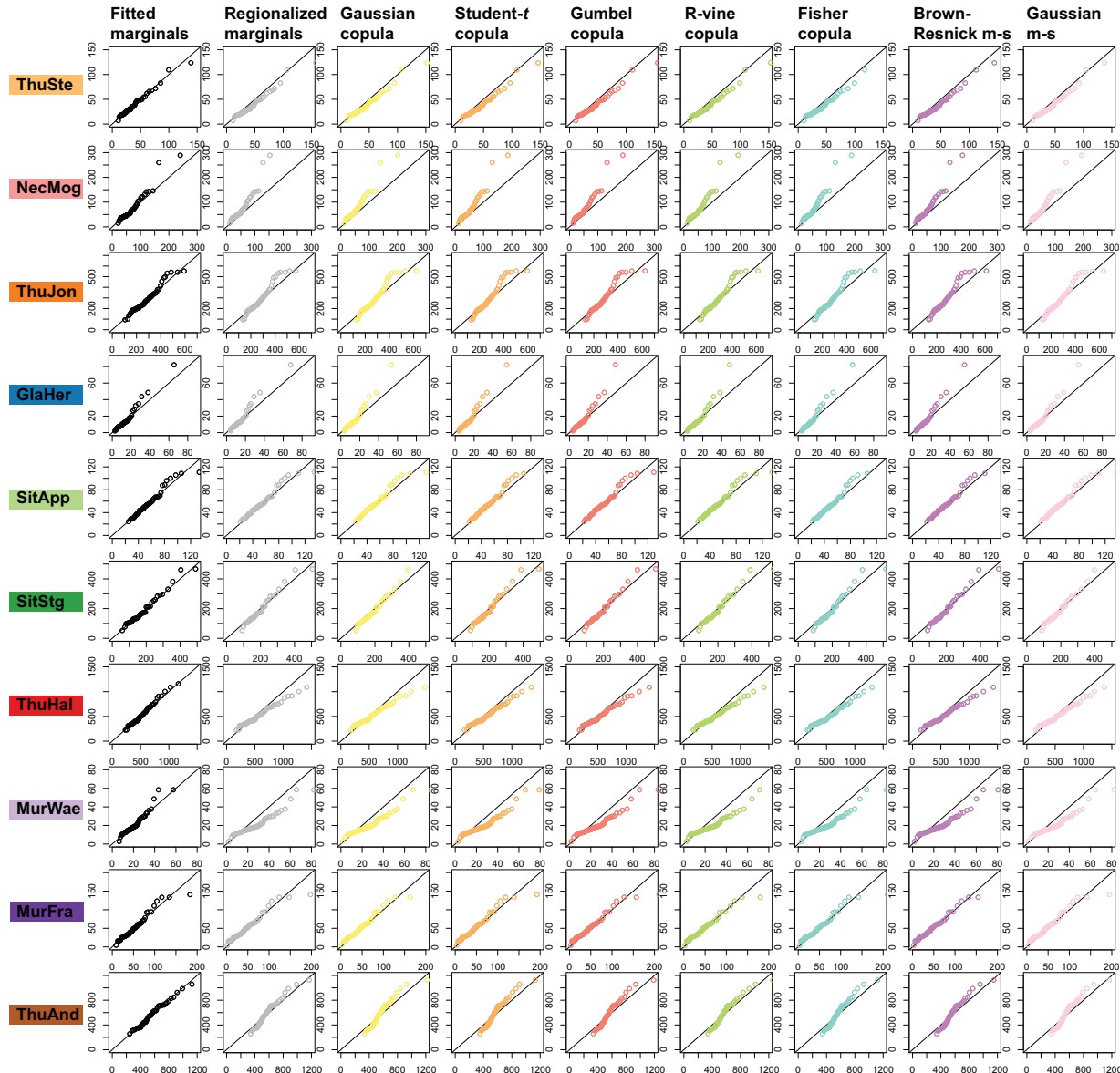

**Figure 6.** The first two columns show QQ-plots obtained by the fitted GEV parameters (Fitted marginals) and by the regionalized GEV parameters (Regionalized marginals) for the ten study stations. The following columns show QQ-plots of samples simulated using different models and back transformed using the regionalized GEV distribution.

**Table 3.** $p$-values for all stations obtained when comparing two samples, observations and simulations, using the Kolmogorov-Smirnov test. The name of the simulated dataset is indicated in the column name. Note that the $p$-values are not adjusted for multiple testing.

| Station | Fitted marginals | Regionalized marginals | Gaussian copula | Student-t copula | Gumbel copula | R-vine copula | Fisher copula | Brown-Resnick m-s | Gaussian m-s |
|---|---|---|---|---|---|---|---|---|---|
| ThuSte | 0.97 | 0.13 | 0.20 | 0.08 | 0.20 | 0.13 | 0.13 | 0.08 | 0.13 |
| NecMog | 0.69 | 0.00 | 0.00 | 0.00 | 0.00 | 0.00 | 0.00 | 0.00 | 0.00 |
| ThuJon | 0.94 | 0.94 | 0.99 | 0.94 | 0.94 | 0.94 | 0.94 | 0.94 | 0.94 |
| GlaHer | 0.94 | 0.54 | 0.83 | 0.69 | 0.69 | 0.54 | 0.69 | 0.54 | 0.69 |
| SitApp | 1.00 | 0.79 | 0.49 | 0.64 | 0.49 | 0.64 | 0.64 | 0.79 | 0.49 |
| SitStg | 0.96 | 0.37 | 0.37 | 0.25 | 0.52 | 0.37 | 0.37 | 0.37 | 0.52 |
| ThuHal | 0.54 | 0.83 | 0.83 | 0.69 | 0.69 | 0.83 | 0.69 | 0.69 | 0.83 |
| MurWae | 0.83 | 0.01 | 0.01 | 0.01 | 0.02 | 0.01 | 0.01 | 0.01 | 0.02 |
| MurFra | 0.83 | 0.54 | 0.41 | 0.29 | 0.29 | 0.41 | 0.41 | 0.29 | 0.29 |
| ThuAnd | 0.99 | 0.06 | 0.06 | 0.06 | 0.06 | 0.09 | 0.06 | 0.06 | 0.06 |

dependence across pairs of stations and most closely reproduced the tail dependence of the observations even though it also led to an overestimation of tail dependence.

The lower tail dependence in the data, which was rather weak for most pairs of stations, was not reproduced by any of the models tested. It was overestimated by the Student-$t$ copula and the Gaussian max-stable model and underestimated by all other models.

### 4.4 Simulation for gauged locations

Figure 10 displays the return periods of peak discharges simulated at all stations corresponding to the 100-year event at a reference station. The boxplots indicate the range of return periods obtained by repeating the simulation 2000 times. The first subplot of Figure 10 for example chose the subcatchment Thur-Stein as the reference station. After having generated a stochastic event set using the Fisher copula, the 100-year event at this station was determined using univariate frequency analysis. Then, the return period of the values corresponding to the same generated event set at the other stations was determined by looking at their univariate, empirical distributions. The simulation procedure was repeated 2000 times which resulted in a range of return periods for each station corresponding to the 100-year event at the reference station. A 100 year event at one station was usually associated with events with return periods higher than 100 years at some stations and with events with lower return periods than 100 years at other stations. Some stations were rather independent from others since their return periods differed strongly from the return periods of the same events at other stations. The values generated for the tributaries of the

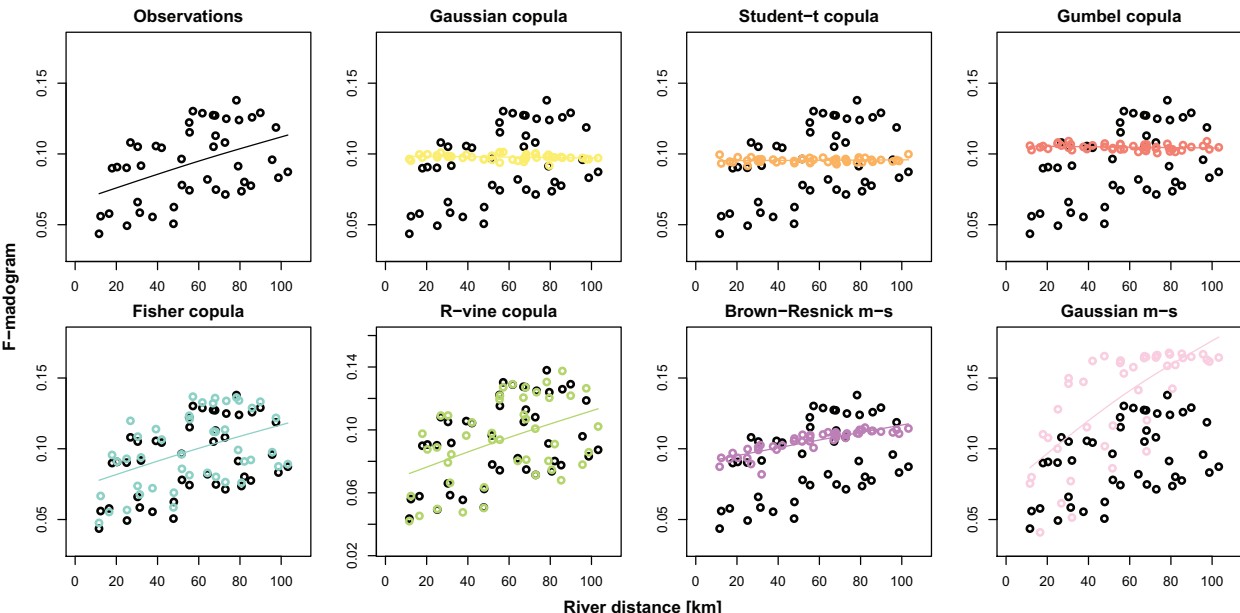

**Figure 7.** F-madograms of observations and samples simulated by different types of dependence models. A smoothed line was added to the data. Each simulated dataset is displayed in a separate panel to increase readability.

Murg and the Glatt could have rather low return periods while the Thur at Andelfingen could show high return periods and vice versa (see subplots GlaHer, MurFra, and ThuAnd).

## 4.5 Interpolation to ungauged location(s)

The Fisher copula was applied in a spatial context to generate event samples for a set of locations consisting of both gauged

5 and ungauged locations. The dependence structure of a sample generated using the Fisher copula is shown in Figure 11. The dependence structure among the gauged catchments was similar to the one obtained when using the model consisting of the gauged stations only. The dependence structure obtained for the ungauged catchments could not be directly compared to observations but it could be verified by comparing the dependence values to the locations of the stations on the river network (Figure 1). Stations close to one another in terms of river distance should have a higher dependence than stations which are

10 not part of the same sub-catchment and separated by longer river distances. The gauged station Thur-Halden and the ungauged location 3 for example showed high dependence in terms of Kendall's tau since they are very close to each other. The same could be observed for the ungauged locations 1 and 10. In contrast, the gauged station Thur-Stein and the ungauged location 8 showed a low dependence because they are separated by a long river distance even though they are quite close in terms of Euclidean distance. The same was also true for e.g. the ungauged locations 8 and 11.

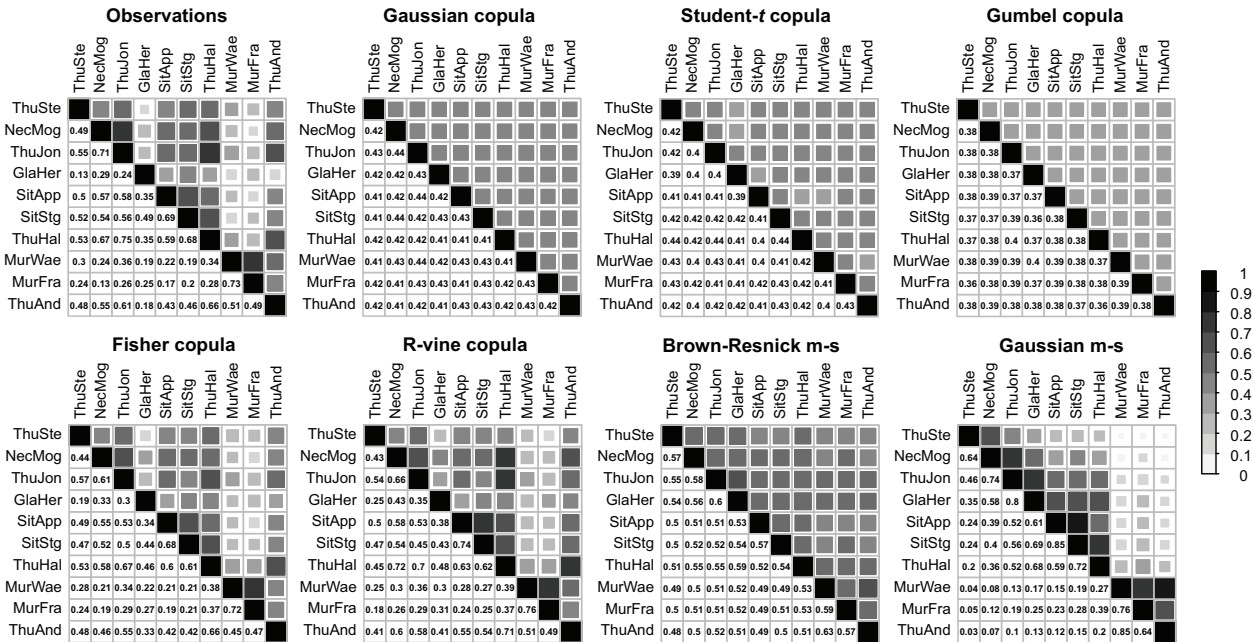

**Figure 8.** Kendall's tau matrices for the observations and the event samples generated by different types of dependence models. The darker the color, the stronger the dependence.

## 5 Discussion

The Euclidean distance is used as a standard distance measure when looking at spatial extremes such as precipitation, temperature, and wind (Asadi et al., 2015). However, our study as well as the one by Asadi et al. (2015) for the Danube river has shown that the Euclidean distance between stations has low explanatory power in the case of regional floods. We found that river distance is a suitable dependence measure for describing the change in dependence along the river network. Other distances such as distance with respect to mean altitude were also found to be suitable since it is strongly related to distance along the river network. Our results have shown that correlations and the F-madogram between stations at the same river distance can still vary quite a bit since two pairs of stations could be located in the upper or lower part of the catchment which seems to have an impact on the correlation coefficient. This clearly shows the need for a multivariate assessment of dependence.

Most models tested were not able to reproduce the dependence structure in the flood event data of the Thur catchment with respect to the river distance between the individual locations. This was indicated by the non-variable F-madogram estimates for the Gaussian, Student-t, and Gumbel copulas. The Gaussian and Student-$t$ copulas were not able to reproduce the dependence structure in the data because they have symmetric tail dependence and were not able to model the decrease in dependence with increasing distance between the stations. This is in line with results by Davison et al. (2012) who found that the use of Gaussian

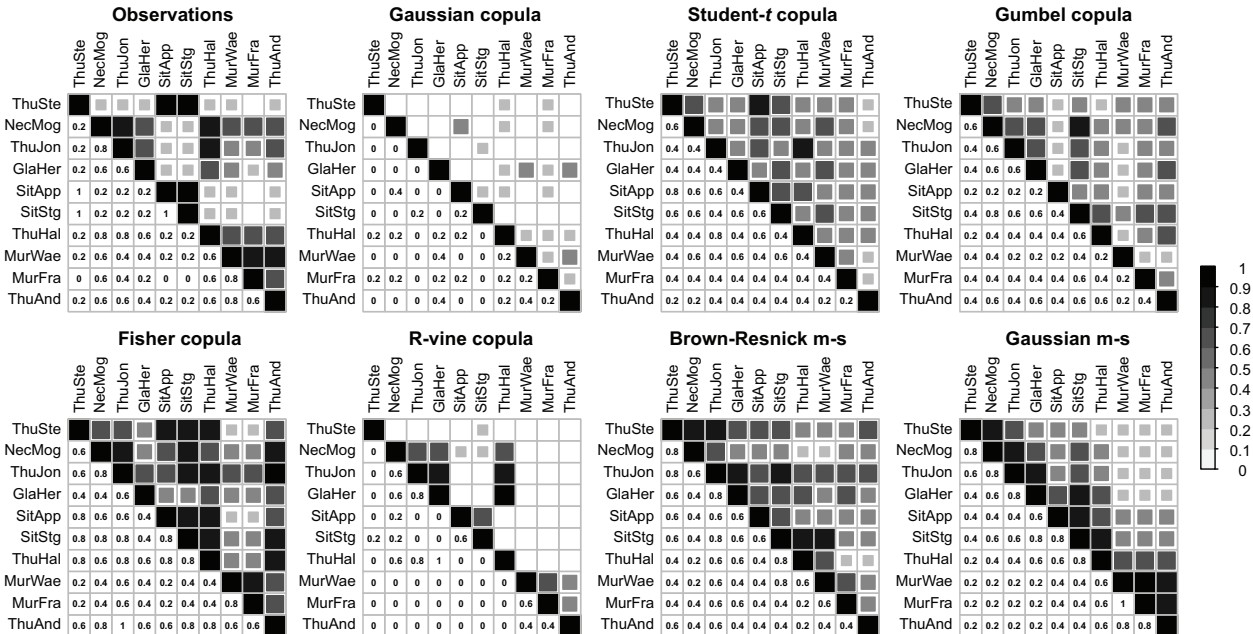

**Figure 9.** Upper tail dependence matrices for the observations and the event samples generated by different types of dependence models. The darker the color, the stronger the dependence.

copulas was not satisfactory when looking at extreme precipitation. Similarly, the Gumbel copula was not able to model the dependence structure in the data despite its asymmetry in tail dependence. The max-stable models did also neither capture the general dependence structure in the data nor the upper or lower tail dependence. The F-madogram produced by the Gaussian max-stable process was more variable than the one of the Brown-Resnick max-stable process, which better fitted the observed data. The difference between the two max-stable models might be explained by the higher flexibility of the Brown-Resnick process compared to the Gaussian process (Cooley et al., 2012). The R-vine copula was thanks to its flexible structure able to reproduce the general dependence structure. This is in line with findings by Schulte and Schumann (2015) who worked with pair copula constructions and found that the use of flexible Archimedean copulas allowed for representing spatial dependence in flood event data. This is essentially due to the fact that Archimedean copulas can model asymmetric lower and upper tail dependence but they are only available for lower dimensions. In our case, the tail dependence was not well reproduced by the R-vine copulas but as mentioned previously, currently available tail dependence estimators are not very reliable (Serinaldi et al., 2015) and all the results regarding tail dependence need to be interpreted with care.

The Fisher copula in contrast to most other models tested well reproduced the general dependence structure in the data as described by Kendall's tau. This was possible since the empirical correlation matrix was used for computations and there exist

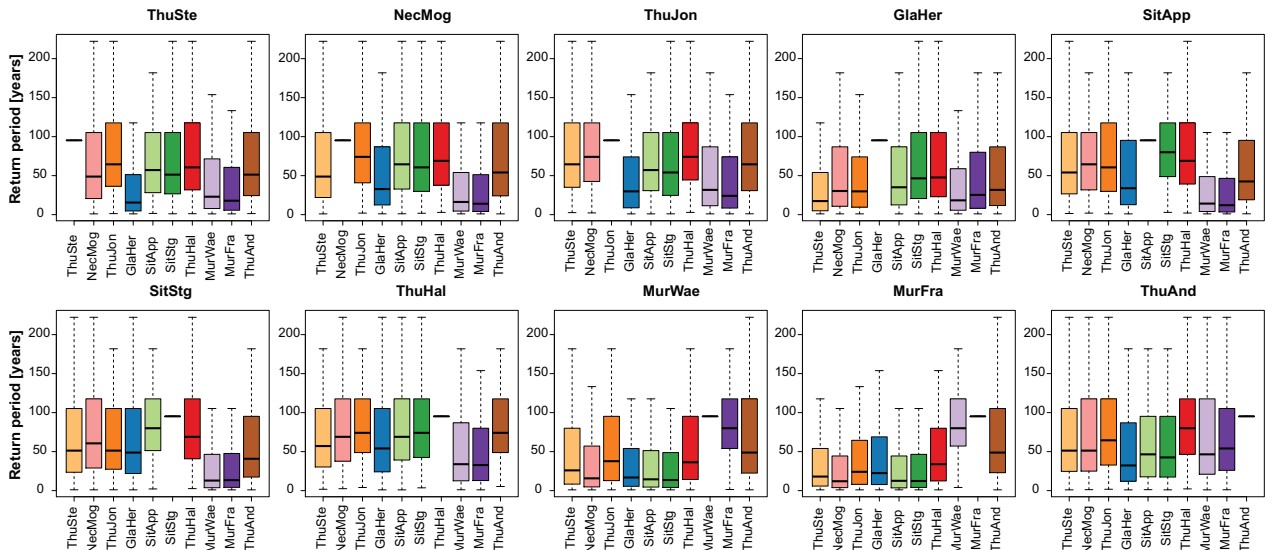

**Figure 10.** Return period of the events of all stations corresponding to the 100-year flood event determined at one single reference station. The reference station is indicated by the horizontal tick at a return period of 100 years. The interquartile range and the whiskers (last value within 1.5 times the interquartile range) indicate the range of the return periods of the values corresponding to the same event set measured at the other stations.

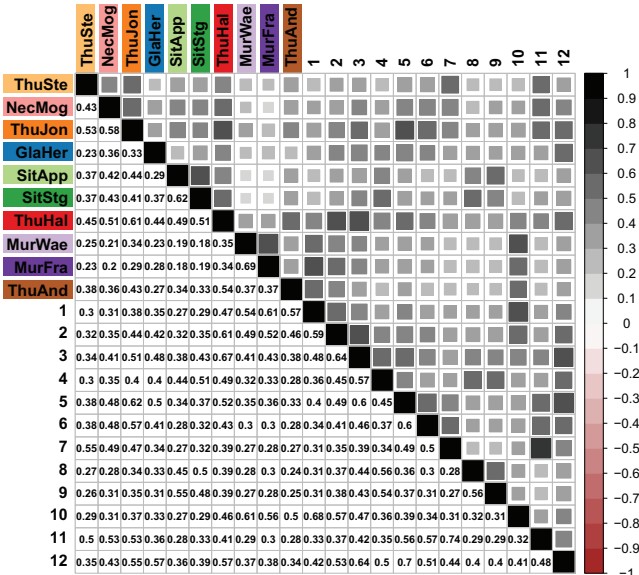

**Figure 11.** Kendall's tau matrix of samples generated by the Fisher copula for the 10 gauged (colored abbreviations) and the 12 ungauged locations (numbers 1-12) (see Figure 1).The Kendall's tau matrix of the samples generated does not equal the matrix displayed in Figure 8 for the Fisher copula since the samples are randomly generated and therefore not perfectly reproducible.

many degrees of freedom. The Fisher copula was also able to model the upper tail of the distribution while it was unable to reproduce the lower tail dependence structure in the data. This is not surprising since the Fisher copula has a non-null upper tail dependence coefficient but no non-null lower tail dependence coefficient (Favre et al., 2018). If one wanted to give even more weight to the upper tail of the distribution, the inference of the Fisher copula parameter $\nu$ could be based on the upper tail dependence matrix instead of the Kendall's tau matrix. Currently, to our knowledge, no copula model in more than three dimensions is available which models asymmetric lower and upper tail dependence. Such a class of copulas could be developed by squaring the components of elliptical random vectors other than the Student distribution (e.g. Laplace or logistic). One copula potentially able to model asymmetric lower and upper tail dependence might be the copula developed by Khoudraji (1995) and generalized by Durante and Salvadori (2010) to a $d$-dimensional space. However, the tail dependence of this type of copulas has not been studied yet. Ghizzoni et al. (2010) and Ghizzoni et al. (2012) have shown that the multivariate version of the skew-$t$ distribution can be an alternative to model spatial dependence in flood events since it allows for asymmetric upper and lower tail dependence and can be used in high dimensions. However, this approach has not yet been extended to an interpolation context where event sets can be simulated for both gauged and ungauged locations. Another alternative for flexibly modeling the spatial dependence in the data is the conditional exceedance model by Heffernan and Tawn (2004) which has successfully been applied in the context of flood events by Keef et al. (2013) and Neal et al. (2013). This model accounts for varying strengths of extremal dependence, however, a number of models must be fit conditioning on each gauging station in turn and no interpolation to ungauged locations is possible so far.

On the contrary to the conditional exceedance and the multivariate skew-$t$ distribution approaches, the Fisher copula can be applied for the simulation of event sets including ungauged locations. The Fisher copula was adjusted by extending the $\Sigma$ correlation matrix to the desired dimension using the values of a fitted correlogram corresponding to the river distances among stations. An alternative method to this statistical approach, would be to use a rainfall-runoff model on stochastically generated precipitation series for the generation of event sets including ungauged locations (Haberlandt et al., 2011). Rainfall-runoff modeling implicitly creates spatial dependence between flows at different locations (Neal et al., 2013). However, the use of a rainfall-runoff model compared to a statistical analysis of flow records, introduces additional sources of uncertainties such as model and parameter uncertainty of the hydrological model (Sikorska et al., 2015). Focusing on flood event data avoids complexity and reduces the number of uncertainty sources (Keef et al., 2013). In the approach presented here, uncertainty arises from both the marginal model and the spatial dependence model. The uncertainty coming from these two uncertainty sources would need to be assessed in a subsequent study. Such a study could be set up in a similar way as studies which have been performed in the context of uncertainty of flood estimates characterizing flood events in terms of the dependent variables peak discharge, flood volume, and duration. Brunner et al. (2018b), Dung et al. (2015) and Serinaldi (2013) have shown how the uncertainty due to the marginal distributions and the dependence between variables can jointly be assessed.

In addition to the spatial dependence, the marginal distributions had to be regionalized to ungauged catchments. They were regionalized using a simple linear model which led to an acceptable performance. However, more sophisticated regionalization techniques such as the use of non-linear regression techniques (Brunner et al., 2018a), the use of the region of influence approach (Hosking and Wallis, 1997), or the copula based clustering algorithm proposed by Pappadà et al. (2018) could be applied

if the marginal distributions were of particular interest and if a larger dataset was available. Alternatively, new regionalization techniques based on catchment similarity in terms of empirical copulas (Grimaldi et al., 2016) could be developed. A larger dataset would also be desirable for reliable estimation of the dependence structure. One possibility of increasing the dataset would be to exploit the whole content of a continuous runoff series instead of only using a flood event sample as suggested by

Serinaldi and Kilsby (2017).

So far, the spatial Fisher copula model has only been applied to the Thur study region. However, its application is not limited to this particular catchment. The Fisher copula can potentially be fitted to the flood event data of other regions since it is quite flexible. When transferring the method to other regions one might need to answer the question of whether Alpine and lowland catchments or humid and dry catchments show similar dependence structures or whether the spatial dependence structure

varies with the topography and the hydro-climatology of the catchment. Keef et al. (2009) have shown that spatial dependence is strong in areas where catchment characteristics are similar. A point which has not been addressed here is whether the spatial dependence structure is constant over time. Blanchet et al. (2018) have found that there was a trend in the co-occurrence of extreme precipitation events. Such a trend analysis should ideally also be performed on a flood dataset covering a longer observation period.

**6  Conclusions**

River distance was found to be a suitable distance measure for explaining the spatial dependence in the flood event data of the Thur catchment in Switzerland. In contrast to other copula and max-stable models, the Fisher copula was able to model both the general dependence structure of the data and its upper tail dependence. However, it failed at capturing the lower tail dependence in the data. Thanks to the main interest in the upper tail of the distribution when modeling flood events, it was still

found to be a suitable model for representing the spatial dependence of flood events. Due to its flexibility it can be adjusted to a context where both gauged and ungauged catchments can be modeled. Flood event sets generated by the Fisher copula can be used in spatial flood risk analyses by combining them with hydraulic models and flood loss models.

*Data availability.* The data used is available upon order from the FOEN using the form under https://www.bafu.admin.ch/bafu/en/home/topics/water/state/ monitoring-data-on-the-topic-of-water/hydrological-data-service-for-watercourses-and-lakes.html.

*Author contributions.* The idea and setup for the paper was jointly developed by the three co-authors. The analyses were performed by Manuela Brunner and discussed with the co-authors. Manuela Brunner wrote the first draft of the manuscript which was revised and edited by Reinhard Furrer and Anne-Catherine Favre.

*Competing interests.* The authors do not have any competing interests

*Acknowledgements.* We thank the Federal Office for the Environment (FOEN) for funding the project (contract 13.0028.KP/M285-0623) and for providing runoff measurement data. We also thank Daniel Viviroli for providing the catchment characteristics data for the ten gauged catchments.

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
