# Peer review of "Modeling the spatial dependence of floods using the Fisher copula"

_Hydrology and Earth System Sciences, 2018_

## Referee Comment (RC1) · Anonymous Referee #1 · 3 Jul 2018

**REVIEW REPORT**

**Journal:** Hydrology and Earth System Sciences
**Paper:** hess-2018-159
**Title:** Modeling the spatial dependence of floods using the Fisher copula
**Author(s):** Manuela I. Brunner, Reinhard Furrer, and Anne-Catherine Favre

**GENERAL COMMENTS.**

This is a nice paper: it is clear, well written, it deals with a problem of interest for the readers, and introduces several elements of novelty, which are well combined together in an appropriate way. Therefore, I may anticipate that I am in favour of having this work published. However, a few critical issues must be fixed before acceptance. Below, please find some indications: the objections should be read in a constructive way, since they may help the Authors improve the paper.

As a final note, for the benefit of unskilled readers and practitioners, the Authors should provide some basic and thorough references involving seminal books, papers, and guidelines about copula modeling, like writing:

"For a theoretical introduction to copulas, see Nelsen (2006); Joe (2014); Durante and Sempi (2015); for a practical/engineering approach, see Genest and Favre (2007); Salvadori and De Michele (2007); Salvadori et al. (2007). In particular, elementary Guidelines for using copulas are illustrated in Favre et al. (2004); Salvadori and De Michele (2004); Salvadori et al. (2014) (and references therein) for multivariate frequency analysis and design, and in Salvadori et al. (2015, 2016) for a multivariate structural approach."

**SPECIFIC COMMENTS.**

**Page(s) 1, Line(s) 15–16.**

   **Author(s).** The Fisher copula is therefore a suitable model for the stochastic simulation of flood event sets at multiple gauged and ungauged locations.

   **Referee.** Such a claim is too strong and general, since it is based on a single case study: please make it weaker.

**Page(s) 5, Line(s) 5–6.**

   **Author(s).** First, flood events were identified at a local scale for each individual station using a peak-over-threshold approach with the 0.9975 quantile as a threshold.

   **Referee.** Is this the 99.75% quantile of the empirical distribution of the hourly data collected at each station? Did I get it right?

**Page(s) 5, Line(s) 9–10.**

   **Author(s).** This procedure allowed for the composition of an event set with events during which at least one station exceeded its 0.9975 quantile.

   **Referee.** If I understand it correctly, the Authors use a multivariate "OR" Hazard Scenario approach, as thoroughly conceptually defined and discussed in Salvadori et al. (2016): please make the point clear.

**Page(s) 5, Line(s) 15–16.**

Author(s). Criterion one was a low variability of the ranks of one event across different stations (standard deviation of ranks ¡ 50)...

Referee. The explanation is somewhat obscure, and maybe I did not get it right. Why using the standard deviation of the ranks and not of the observations? The ranks are the same for all stations (i.e., integers from 1 to $N$), and cannot express the actual intensity/magnitude of the phenomenon. To be clear: the observations may have the same ranks at two different stations, but the discharges measured at one station may be, say, 10 times larger than the ones observed at another station. Please make the point clear, and similarly for Criterion 2.

**Page(s) 6, Figure(s) 2.**

In the caption of Figure 2, the Authors may add that patterns of positive association are clearly visible in all cases.

**Page(s) 6, Line(s) 9–10.**

Author(s). It shows that there is a dependence...

Referee. The Authors may add that, in particular, the variables are in general positively associated.

**Page(s) 6, Line(s) 11–13.**

Author(s). Both upper and lower tail dependence were present in the data according to the estimator of Schmidt and Stadtmuller (2006) which needs to be used with care since it provides unreliable estimates for small sample sizes (Serinaldi, 2015).

Referee. The Authors correctly warn the reader about the problems concerning the estimate of the Tail Dependence coefficients. However, later (e.g., at page 19), they write sentences like "this model fits the tail dependence better than the other model". I would suggest to check the manuscript, and change (if not discard) claims like the one mentioned above. In fact, given the uncertainty of the estimates of the Tail Dependence coefficients (if not "randomness" of the estimate, as in the numerical experiments under controlled conditions I personally carried out using the same estimators suggested by the Authors), I think it could be dangerous to use, and to rely on, the notion of Tail Dependence.

**Page(s) 7, Line(s) 4–5.**

Author(s). We used river distance as a distance measure since it has a hydrological meaning.

Referee. This is an interesting point, and may provide a valuable solution: I like it!

**Page(s) 7, Line(s) 14–15.**

Author(s). The generalized extreme value distribution (GEV) (Coles, 2001) was not rejected for both types of events according to the Kolmogorov-Smirnov test statistic (level $\alpha = 0.05$).

Referee. This is a critical statistical point: how was the p-value computed? In fact, as is well known (e.g., simply read the help of Matlab), the KS test requires that the theoretical distribution be known a priori, it cannot be the fitted one. In the latter case, suitable (but simple) Monte Carlo techniques can be used to estimate an approximate p-value. Please clarify the issue.

**Page(s) 7, Line(s) 24–15.**

Why the Y-coordinate is not present in the case of the Location parameter?

**Page(s) 7, Line(s) 29.**

**Author(s).** They resulted in absolute prediction errors over the ten stations of 0.11, 0.21, and 0.15 respectively.

**Referee.** This result is somewhat difficult to interpret from a practical point of view: please provide some explanation.

**Page(s) 9, Line(s) 7–ff.**

**Author(s).** ... the $d$-dimensional Fisher copula can be expressed by...

**Referee.** In the definition of the Fisher copula, it is not clear what the $\epsilon$'s are. The mathematical notation used is confusing (as well as in the cited original paper by Favre et al., 2018). Are these variables/parameters continuous or discrete, as it seems (the braces notation $\{-1, +1\}^d$ does not help)? It is not clear whether they just take the values $-1$ and $+1$, or all values in the subset (open? closed?) $(-1, +1)$. Please clarify the issue. Furthermore, considering a practical perspective, what is the "role/contribution" of the $\epsilon$'s to the dependence structure? How do they affect the copula? Sorry, I am puzzled: a better explanation would help the reader.

**Page(s) 10, Line(s) 2–3.**

**Author(s).** Max-stable processes assume asymptotic dependence (i.e. dependence will not disappear at very large distances)...

**Referee.** This claim is not clear: asymptotic dependence looks like a mathematical (analytical) property: what is the notion of "distance" mentioned by the Authors? Is it a physical/geographical distance? Please make things clear.

**Page(s) 10, Line(s) 12.**

**Author(s).** To get values on the original scale, these values had to be back-transformed.

**Referee.** Practically, the Authors used a Probability Integral Transform procedure, isn't it?

**Page(s) 10, Line(s) 15.**

**Author(s).** The most suitable model was defined as the one that best reproduced the observed data.

**Referee.** This sentence is "void", since no criteria are specified to define the notion of "best reproduced". Please fix this claim.

**Page(s) 10, Line(s) 16.**

**Author(s).** ... no quantitative goodness of fit test is available to help in selecting the best model...

**Referee.** This sentence is (statistically) questionable: a GoF test can only be used to reject a distribution, surely it CANNOT / MUST NOT be used to select a distribution. Please fix this claim.

**Page(s) 11, Line(s) 10–ff.**

Usually, in hydrology, an elementary way to test a procedure for ungauged sites is to check it by using all gauged locations except one, whose records are known (but are supposed to be unknown). It is not clear whether the Authors followed this approach. If not, first the Authors should use this protocol, and exclude, one at a time, each one of the known stations, and check how, and to what extent, the model they propose is able to reproduce the known (but discarded) data.

**Page(s) 13, Line(s) 1–2.**

**Author(s).** The samples generated using the dependence models and back transformed using the regionalized GEV parameters showed a very similar picture to these regionalized marginal distributions.

**Referee.** Is it possible to provide a table of p-values? Visual statistics is intuitive, but some more "objective" results would be better.

**Page(s) 13, Line(s) 4.**

**Author(s).** Figure 6 shows the F-madogram of the observations and the different dependence models.

**Referee.** Figure 6 is confusing. I would suggest to use a 4x2 frame, and show all the plots of individual comparisons: this would make graphically clear the ability of each model to fit (or not) the data.

**Page(s) 17, Figure(s) 8.**

To the best of my knowledge, the Tail Dependence coefficient ranges from 0 to 1: why the colorbar ranges from $-1$ to $+1$? Just a software feature?

**Page(s) 17, Line(s) 13–14.**

**Author(s).** Similarly, the Gumbel copula was not able to model the dependence structure in the data despite its asymmetry.

**Referee.** The Authors should make the claim more precise. The Gumbel copula is symmetric, being Archimedean: the asymmetry concerns the tail dependence (only upper, not lower). Please fix the sentence.

**Page(s) 19, Line(s) 15–16.**

**Author(s).** Currently, to our knowledge, no copula model in more than three dimensions is available which models asymmetric lower and upper tail dependence.

**Referee.** Intuitively, this should be possible by using the Khoudraji-Liebscher copulas introduced by Durante and Salvadori (2010), but I did not check it.

**Page(s) 20, Line(s) 5–6.**

**Author(s).** However, more sophisticated regionalization techniques such as. . .

**Referee.** The Authors may also consider mentioning recent regionalization approaches entirely based on copulas, such as the ones outlined in Grimaldi et al. (2016) and Pappadà et al. (2018).

**References**

Durante, F., Salvadori, G., 2010. On the construction of multivariate extreme value models via copulas. Environmetrics 21, 143–161.

Durante, F., Sempi, C., 2015. Principles of copula theory. CRC/Chapman & Hall, Boca Raton, FL.

Favre, A.-C., El Adlouni, S., Perreault, L., Thiémonge, N., Bobée, B., 2004. Multivariate hydrological frequency analysis using copulas. Water Resources Research 40 (1).

Genest, C., Favre, A., 2007. Everything you always wanted to know about copula modeling but were afraid to ask. Journal of Hydrologic Engineering 12 (4), 347–368.

Grimaldi, S., Petroselli, A., Salvadori, G., De Michele, C., 2016. Catchment compatibility via copulas: A non-parametric study of the dependence structures of hydrological responses. Advances in Water Resources 90, 116 – 133, doi: 10.1016/j.advwatres.2016.02.003.

Joe, H., 2014. Dependence Modeling with Copulas. CRC Monographs on Statistics & Applied Probability. Chapman & Hall, London.

Nelsen, R., 2006. An introduction to copulas, 2nd Edition. Springer-Verlag, New York.

Pappadà, R., Durante, F., Salvadori, G., De Michele, C., 2018. Clustering of concurrent flood risks via hazard scenarios. Spatial Statistics 23, 124–142, doi: 10.1016/j.spasta.2017.12.002.

Salvadori, G., De Michele, C., 2004. Frequency analysis via copulas: theoretical aspects and applications to hydrological events. Water Resour. Res. 40, W12511, doi: 10.1029/2004WR003133.

Salvadori, G., De Michele, C., 2007. On the use of copulas in hydrology: theory and practice. J. Hydrol. Eng. 12 (4), 369–380, (Special Issue: Copulas in Hydrology; doi: 10.1061/(ASCE)1084-0699(2007)12:4(369)).

Salvadori, G., De Michele, C., Kottegoda, N., Rosso, R., 2007. Extremes in Nature. An approach using Copulas. Vol. 56 of Water Science and Technology Library Series. Springer, Dordrecht, ISBN: 978-1-4020-4415-1.

Salvadori, G., Durante, F., De Michele, C., Bernardi, M., Petrella, L., 2016. A multivariate Copula-based framework for dealing with Hazard Scenarios and Failure Probabilities. Water Resources Research 52 (5), 3701–3721, doi: 10.1002/2015WR017225.

Salvadori, G., Durante, F., Tomasicchio, G. R., D'Alessandro, F., 2015. Practical guidelines for the multivariate assessment of the structural risk in coastal and off-shore engineering. Coastal Engineering 95, 77–83, doi: 10.1016/j.coastaleng.2014.09.007.

Salvadori, G., Tomasicchio, G. R., D'Alessandro, F., 2014. Practical guidelines for multivariate analysis and design in coastal and off-shore engineering. Coastal Engineering 88, 1–14, doi: 10.1016/j.coastaleng.2014.01.011.

---

## Author Comment (AC1) · 28 Aug 2018

**Journal:** Hydrology and Earth System Sciences

**Title:** Modeling the spatial dependence of floods using the Fisher copula.

**Manuscript Number:** hess-2018-159

**Authors:** Manuela I. Brunner, Reinhard Furrer, and Anne-Catherine Favre

**Dear Prof. Bernhardt, dear reviewer,**

We thank the reviewer for acknowledging the value of our work and his/her feedback. We have taken into account his/her suggestions and we would like to answer to his/her comments. We have not yet uploaded the updated version of our manuscript but will do so as soon as it has been revised according to a second reviewer's comments.

Below, we describe step by step which comments have led to which changes in the revised manuscript. Our replies to the reviewers' comments are written in blue and italic to distinct them from the reviewers' comments.

Thank you for your efforts with our manuscript. We hope that you will find the revised version suitable for publication in *Hydrology and Earth System Sciences*.

On the behalf of all co-authors,

Yours sincerely,

Manuela Brunner

**Reviewer 1:**

**GENERAL COMMENTS.**

This is a nice paper: it is clear, well written, it deals with a problem of interest for the readers, and introduces several elements of novelty, which are well combined together in an appropriate way. Therefore, I may anticipate that I am in favour of having this work published. However, a few critical issues must be fixed before acceptance. Below, please find some indications: the objections should be read in a constructive way, since they may help the Authors improve the paper. As a final note, for the benefit of unskilled readers and practitioners, the Authors should provide some basic and thorough references involving seminal books, papers, and guidelines about copula modeling, like writing: "For a theoretical introduction to copulas, see Nelsen (2006); Joe (2014); Durante and Sempi (2015); for a practical/engineering approach, see Genest and Favre (2007); Salvadori and De Michele (2007); Salvadori et al. (2007). In particular, elementary Guidelines for using copulas are illustrated in Favre et al. (2004); Salvadori and De Michele (2004); Salvadori et al. (2014) (and references therein) for multivariate frequency analysis and design, and in Salvadori et al. (2015, 2016) for a multivariate structural approach."

*Reply: Thank you for pointing out that more basic references on copula theory should be provided. We have added the references suggested by the reviewer.*

**SPECIFIC COMMENTS.**

**Page(s) 1, Line(s) 15–16. Author(s).** The Fisher copula is therefore a suitable model for the stochastic simulation of flood event sets at multiple gauged and ungauged locations.

**Referee.** Such a claim is too strong and general, since it is based on a single case study: please make it weaker.

*Reply: We have weakened the statement.*

**Page(s) 5, Line(s) 5–6. Author(s).** First, flood events were identified at a local scale for each individual station using a peak-over-threshold approach with the 0.9975 quantile as a threshold.

**Referee.** Is this the 99.75% quantile of the empirical distribution of the hourly data collected at each station? Did I get it right?

*Reply: Yes, this is correct. We have clarified this in the text.*

**Page(s) 5, Line(s) 9–10. Author(s).** This procedure allowed for the composition of an event set with events during which at least one station exceeded its 0.9975 quantile.

**Referee.** If I understand it correctly, the Authors use a multivariate "OR" Hazard Scenario approach, as thoroughly conceptually defined and discussed in Salvadori et al. (2016): please make the point clear.

*Reply: This is correct and was specified in the text.*

**Page(s) 5, Line(s) 15–16. Author(s).** Criterion one was a low variability of the ranks of one event across different stations (standard deviation of ranks ¡ 50). . .

**Referee.** The explanation is somewhat obscure, and maybe I did not get it right. Why using the standard deviation of the ranks and not of the observations? The ranks are the same for all stations (i.e., integers from 1 to N), and cannot express the actual intensity/magnitude of the phenomenon. To be clear: the observations may have the same ranks at two different stations, but the discharges measured at one station may be, say, 10 times larger than the ones observed at another station. Please make the point clear, and similarly for Criterion 2.

*Reply: Thank you for pointing out that our explanations were not sufficient for a good understanding. We worked with ranks instead of observations to enable a comparison of values measured in catchments with different sizes and therefore event magnitudes. We specified in the text that an event could have a low rank in one series but a high rank in another series, which would lead to a high variability in ranks across stations. On the contrary, an individual event could be assigned similar ranks at different stations, which would lead to a low variability in ranks across stations.*

**Page(s) 6, Figure(s) 2.** In the caption of Figure 2, the Authors may add that patterns of positive association are clearly visible in all cases.

*Reply: This sentence was added to the caption.*

**Page(s) 6, Line(s) 9–10. Author(s).** It shows that there is a dependence. . .

**Referee.** The Authors may add that, in particular, the variables are in general positively associated.

*Reply: This was specified.*

**Page(s) 6, Line(s) 11–13. Author(s).** Both upper and lower tail dependence were present in the data according to the estimator of Schmidt and Stadtmuller (2006) which needs to be used with care since it provides unreliable estimates for small sample sizes (Serinaldi, 2015).

**Referee.** The Authors correctly warn the reader about the problems concerning the estimate of the Tail Dependence coefficients. However, later (e.g., at page 19), they write sentences like "this model fits the tail dependence better than the other model". I would suggest to check the manuscript, and

change (if not discard) claims like the one mentioned above. In fact, given the uncertainty of the estimates of the Tail Dependence coefficients (if not "randomness" of the estimate, as in the numerical experiments under controlled conditions I personally carried out using the same estimators suggested by the Authors), I think it could be dangerous to use, and to rely on, the notion of Tail Dependence.

*Reply: Thank you for stressing this important point. We have added a sentence to the Discussion which again points out the deficiencies in currently available tail dependence estimators. We find it generally very difficult how to address this issue in scientific publications. On the one hand, tail dependence is said to be an important statistical property of the data and important for making suitable model choices, but on the other hand, we do neither have sufficiently long data series nor appropriate tools to quantify it.*

**Page(s) 7, Line(s) 4–5. Author(s).** We used river distance as a distance measure since it has a hydrological meaning.

**Referee.** This is an interesting point, and may provide a valuable solution: I like it!

*Reply: Thank you.*

**Page(s) 7, Line(s) 14–15. Author(s).** The generalized extreme value distribution (GEV) (Coles, 2001) was not rejected for both types of events according to the Kolmogorov-Smirnov test statistic (level = 0:05).

**Referee.** This is a critical statistical point: how was the p-value computed? In fact, as is well known (e.g., simply read the help of Matlab), the KS test requires that the theoretical distribution be known a priori, it cannot be the fitted one. In the latter case, suitable (but simple) Monte Carlo techniques can be used to estimate an approximate p-value. Please clarify the issue.

*Reply: Thank you very much for pointing out this critical point. We got aware that using the Kolmogorov-Smirnov test on estimated parameters as done in the previous version of the manuscript was not appropriate. We now use the Anderson-Darling test instead. This test was found to be appropriate for testing the validity of different families of skewed distributions such as the generalized extreme value distribution when unknown parameters have to be estimated from the sample (Chen & Balakrishnan, 1995). This test is implemented in the R-package gnfit by (Saeb, 2018).*

**Page(s) 7, Line(s) 24–15.** Why the Y-coordinate is not present in the case of the Location parameter?

*Reply: We applied stepwise backward regression to identify those explanatory variables which could significantly improve predictions of the location parameter. The Y-coordinate was not found to be one of these predictors.*

**Page(s) 7, Line(s) 29. Author(s).** They resulted in absolute prediction errors over the ten stations of 0.11, 0.21, and 0.15 respectively.

**Referee.** This result is somewhat difficult to interpret from a practical point of view: please provide some explanation.

*Reply: We point out that already the use of the regional marginal model leads to prediction errors. However, the marginal models were not the focus of this study and more sophisticated methods could be thought for improving these errors as mentioned in the Discussion section.*

**Page(s) 9, Line(s) 7–ff. Author(s).** . . . the d-dimensional Fisher copula can be expressed by. . .

**Referee.** In the definition of the Fisher copula, it is not clear what the ε's are. The mathematical notation used is confusing (as well as in the cited original paper by Favre et al., 2018). Are these variables/parameters continuous or discrete, as it seems (the braces notation $\{-1, +1\}^d$ does not help)? It is not clear whether they just take the values -1 and +1, or all values in the subset (open? closed?) (-1; +1). Please clarify the issue. Furthermore, considering a practical perspective, what is the "role/contribution" of the ε's to the dependence structure? How do they affect the copula? Sorry, I am puzzled: a better explanation would help the reader.

*Reply: We agree that a clarification is necessary. {-1,1} means ε takes either the value -1 or 1. This notation is used in order to obtain a compact expression for the copula. This compact expression has been first introduced by Quessy et al. (2016) for the bivariate chi-square copula replacing the mathematical function sign and an absolute value (see equation (5) in Quessy et al. (2016)). Here, as we have a d-dimensional Fisher copula; we use the notation $\{-1, +1\}^d$. The parameter ε enters in the definition of the copula and does not have a particular role with respect to the dependence structure.*

**Page(s) 10, Line(s) 2–3. Author(s).** Max-stable processes assume asymptotic dependence (i.e. dependence will not disappear at very large distances). . .

**Referee.** This claim is not clear: asymptotic dependence looks like a mathematical (analytical) property: what is the notion of "distance" mentioned by the Authors? Is it a physical/geographical distance? Please make things clear.

*Reply: Thank you for pointing out this unclarity. We have specified in the text that with distance, we mean the distance between two stations.*

**Page(s) 10, Line(s) 12. Author(s).** To get values on the original scale, these values had to be back-transformed.

**Referee.** Practically, the Authors used a Probability Integral Transform procedure, isn't it?

*Reply: Yes, this was specified in the text.*

**Page(s) 10, Line(s) 15. Author(s).** The most suitable model was defined as the one that best reproduced the observed data.

**Referee.** This sentence is "void", since no criteria are specified to define the notion of "best reproduced". Please fix this claim.

*Reply: We have removed this sentence and rephrased another sentence in this paragraph.*

**Page(s) 10, Line(s) 16. Author(s).** . . . no quantitative goodness of fit test is available to help in selecting the best model. . .

**Referee.** This sentence is (statistically) questionable: a GoF test can only be used to reject a distribution, surely it CANNOT / MUST NOT be used to select a distribution. Please fix this claim.

*Reply: We agree and have rephrased the sentence.*

**Page(s) 11, Line(s) 10–ff.** Usually, in hydrology, an elementary way to test a procedure for ungauged sites is to check it by using all gauged locations except one, whose records are known (but are supposed to be unknown). It is not clear whether the Authors followed this approach. If not, first the Authors should use this protocol, and exclude, one at a time, each one of the known stations, and check how, and to what extent, the model they propose is able to reproduce the known (but discarded) data.

*Reply: Thank you for suggesting to do leave-one-out cross-validation for validating the simulated correlation structures. We have followed this procedure as suggested and found that the correlation matrices resulting from the simulated data established based on a reduced dataset consisting of nine gauged and one ungauged station were similar to the correlation matrix of the observed data. For completeness, we include these results in the answer to the reviewers. However, we did not include them in the paper since it is already quite long.*

[Figure]

*Figure 1: Kendall's tau matrices of the observations and simulated datasets obtained by fitting a Fisher copula on nine catchments and applying the interpolation procedure to ungauged catchments on the remaining station.*

**Page(s) 13, Line(s) 1–2. Author(s).** The samples generated using the dependence models and back transformed using the regionalized GEV parameters showed a very similar picture to these regionalized marginal distributions.

**Referee.** Is it possible to provide a table of p-values? Visual statistics is intuitive, but some more "objective" results would be better.

*Reply: We have added a table containing the p-values obtained by doing the Kolmogorov-Smirnov test.*

**Page(s) 13, Line(s) 4. Author(s).** Figure 6 shows the F-madogram of the observations and the different dependence models.

**Referee.** Figure 6 is confusing. I would suggest to use a 4x2 frame, and show all the plots of individual comparisons: this would make graphically clear the ability of each model to fit (or not) the data.

*Reply: Thank you for this suggestion. We have adjusted the figure as suggested, which helped to increase readability.*

**Page(s) 17, Figure(s) 8.**

To the best of my knowledge, the Tail Dependence coefficient ranges from 0 to 1: why the colorbar ranges from -1 to +1? Just a software feature?

*Reply: We have fixed the legend.*

**Page(s) 17, Line(s) 13–14. Author(s).** Similarly, the Gumbel copula was not able to model the dependence structure in the data despite its asymmetry.

**Referee.** The Authors should make the claim more precise. The Gumbel copula is symmetric, being Archimedean: the asymmetry concerns the tail dependence (only upper, not lower). Please fix the sentence.

*Reply: Thank you for pointing this out. We have made the specification.*

**Page(s) 19, Line(s) 15–16. Author(s).** Currently, to our knowledge, no copula model in more than three dimensions is available which models asymmetric lower and upper tail dependence.

**Referee.** Intuitively, this should be possible by using the Khoudraji-Liebscher copulas introduced by Durante and Salvadori (2010), but I did not check it.

*Reply: Khoudraji has first developed a family of bivariate asymmetric extreme value copula in his PhD thesis (Khoudraji, 1995). This family of copula is very general and it has been then generalized by Durante & Salvadori (2010) in d-dimensional space. The family depends of two copulas (called A and B in equation (8) of Durante & Salvadori (2010). To our knowledge the tail dependence of these type of copulas have not been computed and studied. We believe that it would be possible to find appropriate A and B copulas in order to model asymmetric tail dependence. This has been added as a perspective.*

**Page(s) 20, Line(s) 5–6. Author(s).** However, more sophisticated regionalization techniques such as...

**Referee.** The Authors may also consider mentioning recent regionalization approaches entirely based on copulas, such as the ones outlined in Grimaldi et al. (2016) and Pappad`a et al. (2018).

*Reply: Thank you for pointing out these approaches which we have added to the Discussion.*

**References provided by the reviewer**

Durante, F., Salvadori, G., 2010. On the construction of multivariate extreme value models via copulas. Environmetrics 21, 143–161.

Durante, F., Sempi, C., 2015. Principles of copula theory. CRC/Chapman & Hall, Boca Raton, FL.

Favre, A.-C., El Adlouni, S., Perreault, L., Thi´emonge, N., Bob´ee, B., 2004. Multivariate hydrological frequency analysis using copulas. Water Resources Research 40 (1).

Genest, C., Favre, A., 2007. Everything you always wanted to know about copula modeling but were afraid to ask. Journal of Hydrologic Engineering 12 (4), 347–368.

Grimaldi, S., Petroselli, A., Salvadori, G., De Michele, C., 2016. Catchment compatibility via copulas: A non-parametric study of the dependence structures of hydrological responses. Advances in Water Resources 90, 116 – 133, doi: 10.1016/j.advwatres.2016.02.003.

Joe, H., 2014. Dependence Modeling with Copulas. CRC Monographs on Statistics & Applied Probability. Chapman & Hall, London.

Nelsen, R., 2006. An introduction to copulas, 2nd Edition. Springer-Verlag, New York.

Pappada, R., Durante, F., Salvadori, G., De Michele, C., 2018. Clustering of concurrent flood risks via hazard scenarios. Spatial Statistics 23, 124–142, doi: 10.1016/j.spasta.2017.12.002.

Salvadori, G., De Michele, C., 2004. Frequency analysis via copulas: theoretical aspects and applications to hydrological events. Water Resour. Res. 40, W12511, doi: 10.1029/2004WR003133.

Salvadori, G., De Michele, C., 2007. On the use of copulas in hydrology: theory and practice. J. Hydrol. Eng. 12 (4), 369–380, (Special Issue: Copulas in Hydrology; doi: 10.1061/(ASCE)1084-0699(2007)12:4(369)).

Salvadori, G., De Michele, C., Kottegoda, N., Rosso, R., 2007. Extremes in Nature. An approach using Copulas. Vol. 56 of Water Science and Technology Library Series. Springer, Dordrecht, ISBN: 978-1- 4020-4415-1.

Salvadori, G., Durante, F., De Michele, C., Bernardi, M., Petrella, L., 2016. A multivariate Copula-based framework for dealing with Hazard Scenarios and Failure Probabilities. Water Resources Research 52 (5), 3701–3721, doi: 10.1002/2015WR017225.

Salvadori, G., Durante, F., Tomasicchio, G. R., D'Alessandro, F., 2015. Practical guidelines for the multivariate assessment of the structural risk in coastal and off-shore engineering. Coastal Engineering 95, 77–83, doi: 10.1016/j.coastaleng.2014.09.007.

Salvadori, G., Tomasicchio, G. R., D'Alessandro, F., 2014. Practical guidelines for multivariate analysis and design in coastal and off-shore engineering. Coastal Engineering 88, 1–14, doi: 10.1016/j.coastaleng.2014.01.011.

**References used in the response to the reviewer**

Chen, G., & Balakrishnan, N. (1995). A general purpose approximate goodness-of-fit test. *Journal of Quality Technology*, *2*, 154–161.

Durante, F., & Salvadori, G. (2010). On the construction of multivariate extreme value models via copulas. *Environmetrics*, *21*, 143–161.

Khoudraji, A. (1995). *Contribution à l'étude des copules et à la modélisation de valeurs extrêmes bivariées*. Université Laval, Québec, Canada.

Quessy, J. F., Rivest, L. P., & Toupin, M. H. (2016). On the family of multivariate chi-square copulas. *Journal of Multivariate Analysis*, *152*, 40–60. https://doi.org/10.1016/j.jmva.2016.07.007

Saeb, A. (2018). *Package ' gnFit .'*

---

## Referee Comment (RC2) · Anonymous Referee #2 · 20 Nov 2018

The present work analyzes the occurrence of flooding events at a regional scale. In this context, it is crucial to account for the dependence among flowrates at diverse locations. The modelling of the dependence of flooding events is here tackled via the recently proposed Fisher copula. Furthermore, the Authors provide a framework to spatially interpolate flowrate (during the realization of flooding events) at ungauged stations, leveraging on the Fisher copula embedding the dependence among gauged stations. As physical distance for the interpolation of the flowrate, the Authors find that the river length is the most appropriated one. The Authors compare the proposed approach with several others type of Copula and Max-Stable processes, finding that for the Thur catchment in Switzerland the Fisher copula give the best results. I think that the paper is worth for publication after some minor revisions. Comment 1 The

Event Definition procedure could be hard to follow, perhaps a graphical depiction of the procedure (based only on two records and up to line 12 of pp. 5) could help the reader to follow it properly. Comment 2 Figure 3: I would not add the interpolation line connecting the observations, I think it would make more convincing the smoothing spline and the exponential fitting. Comment 3 Figure 6: it is really hard to read the figure. I suggest to split in (a) and (b) panels, where poorly performing models and satisfactory models results are depicted respectively. Comment 4 At first, inspection of Fig. 3 and Fig. 6 suggest a very erratic structure of the dependence (via correlation coefficient and/or F-madogram) in the data as a function of the river distance. This makes suspicious the use of such a distance as a good explanatory variable and the reliability of the current work. Then I have realized that dependence metrics in both Fig.s 4 and 6 are evaluated just among pairs of gauged stations (e.g., the erratic behaviour of the correlation between two stations with a distance of approximately 80Km is due to the fact that the two pairs of stations could be placed in the upper or lower portion of the catchment, making a marked impact on the correlation coefficient. The same for the F-madogram) : this clearly reveals the need for a multivariate assessment of the dependence to me. I think a sentence pointing out this aspect in the text would help the readers.

Please also note the supplement to this comment: https://www.hydrol-earth-syst-sci-discuss.net/hess-2018-159/hess-2018-159-RC2supplement.pdf

---

## Author Comment (AC2) · 23 Nov 2018

**Journal:** Hydrology and Earth System Sciences

**Title:** Modeling the spatial dependence of floods using the Fisher copula.

**Manuscript Number:** hess-2018-159

**Authors:** Manuela I. Brunner, Reinhard Furrer, and Anne-Catherine Favre

**Dear Prof. Guadagnini,**

We thank the reviewer for acknowledging the value of our work and his/her feedback. We have taken into account his/her suggestions and we would like to answer to his/her comments. Below, we describe step by step which comments have led to which changes in the revised manuscript. Our replies to the reviewers' comments are written in blue and italic to distinct them from the reviewers' comments.

Thank you for your efforts with our manuscript. We hope that you will find the revised version suitable for publication in *Hydrology and Earth System Sciences*.

On the behalf of all co-authors,

Yours sincerely,

Manuela Brunner

**Reviewer 2:**

The present work analyzes the occurrence of flooding events at a regional scale. In this context it is crucial to account for the dependence among flowrates at diverse locations. The modelling of the dependence for flooding events is here tackled via the recently proposed Fisher copula. Furthermore, the Authors provide a framework to spatially interpolate flowrate (during the realization of a flooding events) at ungauged stations, leveraging on the Fisher copula embedding the dependence among gauged stations. As physical distance for the interpolation of the flowrate the Authors find that the river length is the most appropriated one. The Authors compare the proposed approach with several others type of Copula and Max-Stable processes, finding that for the Thur catchment in Switzerland the Fisher copula give the best results.
I think that the paper is worth for publication after some minor revisions.

**Comment 1**
The Event Definition procedure could be hard to follow, perhaps a graphical depiction of the procedure (based only on two records and up to line 12 of pp. 5) could help the reader to follow it properly.

***Reply:*** *Thank you for this suggestion. We have added an illustration of the three-step flood event identification procedure in order to be more pedagogical.*

**Comment 2**
Figure 3: I would not add the interpolation line connecting the observations, I think it would make more convincing the smoothing spline and the exponential fitting.

***Reply:*** *We have removed the interpolation line which indeed makes the plot easier to read.*

**Comment 3**

Figure 6: it is really hard to read the figure. I suggest to split in (a) and (b) panels, where poorly performing models and satisfactory models results are depicted respectively.

*Reply: Thank you for pointing this out. This issue was also risen by reviewer 1 who had suggested to plot each model in a separate panel. We did so which largely increased the readability of the plot.*

**Comment 4**

At first, inspection of Fig. 3 and Fig. 6 suggest a very erratic structure of the dependence (via correlation coefficient and/or F-madogram) in the data as a function of the river distance. This make suspicious the use of such a distance as a good explanatory variable and the reliability of the current work. Then I have realized that dependence metrics in both Fig.s 4 and 6 are evaluated just among pairs of gauged stations (e.g., the erratic behavior of the correlation between two stations with a distance of approximately 80Km is due to the fact that the two pairs of stations could be placed in the upper or lower portion of the catchment, making a marked impact on the correlation coefficient. The same for the F-madogram) : this clearly reveals the need for a multivariate assessment of the dependence to me. I think a sentence pointing out this aspect in the text would help the readers.

*Reply: This point was taken into account and added to the discussion section.*